# Vaccination with *Aedes aegypti* AgBR1 Delays Lethal Mosquito-Borne Zika Virus Infection in Mice

**DOI:** 10.3390/vaccines8020145

**Published:** 2020-03-25

**Authors:** Yuchen Wang, Alejandro Marin-Lopez, Junjun Jiang, Michel Ledizet, Erol Fikrig

**Affiliations:** 1State Key Laboratory of Virology, College of Life Science, Wuhan University, Wuhan 430072, China; 2Section of Infectious Diseases, Department of Internal Medicine, Yale University School of Medicine, New Haven, CT 06420, USA; junjun.jiang@yale.edu (J.J.); erol.fikrig@yale.edu (E.F.); 3School of Public Health, Guangxi Medical University, Nanning 530021, China; 4L2 Diagnostics LLC, New Haven, CT 06511, USA; mledizet@L2dx.com; 5Howard Hughes Medical Institute, Chevy Chase, MD 20815, USA

**Keywords:** Zika virus, vaccine, *Aedes aegypti* mosquito, salivary protein

## Abstract

Zika Virus (ZIKV) is transmitted primarily by *Aedes aegypti* mosquitoes, resulting in asymptomatic infection, or acute illness with a fever and headache, or neurological complications, such as Guillain-Barre syndrome or fetal microcephaly. Previously, we determined that AgBR1, a mosquito salivary protein, induces inflammatory responses at the bite site, and that passive immunization with AgBR1 antiserum influences mosquito-transmitted ZIKV infection. Here, we show that the active immunization of mice with AgBR1 adjuvanted with aluminum hydroxide delays lethal mosquito-borne ZIKV infection, suggesting that AgBR1 may be used as part of a vaccine to combat ZIKV.

## 1. Introduction

Zika virus (ZIKV) is a flavivirus primarily spread by *A. aegypti* mosquitoes and causes a range of symptoms, including fever, joint pain, and headaches [1]. In addition, vertical (mother-to-fetus) transmission of the virus may result in microcephaly and other birth defects [2]. There is currently no available licensed vaccine, and most vaccine strategies directly target components of the virus.

Vaccines are often composed of an antigen and an adjuvant that triggers a heightened immunostimulatory response [3]. Alum Hydrogel (Aluminum hydroxide in a wet gel suspension, AH, InvivoGen, CA) is a human-licensed adjuvant that initiates an innate immune response by the enhancement of antigen availability, the activation of antigen presenting cells (APCs), and uptake by immune cells [4]. Aluminum adjuvants have a solid safety record based on more than 70 years of usage [5]. Conventional vaccines in clinical use consist of inactivated or attenuated pathogens or purified pathogen components [6]. For arthropod-borne diseases, infection-enhancing vector molecules co-inoculated with the pathogen may be additional vaccine targets [7]. As the global ZIKV disease burden increases, investigation is warranted of novel vaccines that target mosquito salivary proteins. 

Mosquito salivary proteins enhance the infectivity and pathogenesis of Zika, dengue, or West Nile viruses by modulating immune responses [7], and an antibody response to specific salivary gland proteins influences the pathogenesis of flaviviruses [8,9,10]. Previous studies, including some from our group, have shown that proteins in the saliva of the *A. aegypti* mosquito are capable of potentiating viral infection by arboviruses, demonstrating that certain salivary proteins are important for flavivirus pathogenicity and transmission from vector to host [11,12,13]. Recently, we reported that the *A. aegypti* salivary gland protein AgBR1 modulates the early immune response in the murine skin following mosquito bite. Moreover, passive immunization with antibodies against AgBR1 prolongs survival in mice infected with mosquito-transmitted ZIKV [14]. However, there are limitations to passive immunizations, such as the short-lived protection, the need for repeated administration of antibodies, and the high cost [15]. To determine whether an active immunization strategy could inhibit mosquito-borne ZIKV infection in mice, we adjuvanted AgBR1 with aluminum hydroxide to enhance antibody production and examine its influence on infection. 

## 2. Materials and Methods 

### 2.1. Ethics Statement 

All experiments were performed in accordance with guidelines from the Guide for the Care and Use of Laboratory Animals (National Institutes of Health). The animal experimental protocols were approved by the Institutional Animal Care and Use Committee (IACUC) at the Yale University School of Medicine (assurance number A3230-01). All infection experiments were performed in an arthropod containment level 3 (ACL3) animal facility according to the regulations of Yale University. Every effort was made to minimize murine pain and distress. The mice were anaesthetized with ketamine-xylazine for mosquito infection experiments and euthanized as suggested by the Yale IACUC.

### 2.2. Viruses and Cell Lines

The viruses were propagated in *Aedes aegypti* C6/36 cells grown in Dulbecco modified Eagle medium (DMEM) supplemented with 10% fetal bovine serum (FBS), and 1% penicillin / streptomycin (Invitrogen) at 30°C with 5% CO2. An Asian-derived Cambodian strain (FSS13025 strain, referred to ZIKV^Cam^) isolated in 2010 was propagated in C6/36 cells. 

### 2.3. Mosquitoes and Animals

*Aedes aegypti* (Ho Chi Minh strain, kindly provided by Dr J. Powell laboratory at Yale University) mosquitoes were maintained on 10% sucrose feeders inside a 12- by 12- by 12-in. metal mesh cage (BioQuip; catalog no. 1450B) at 28 °C and 80% humidity. Egg masses were generated via blood meal feeding on naive mice. All mosquitoes were housed in a warm chamber in a space approved for Biosafety Level 2 and Arthropod Containment Level 3 research. Mosquitoes were randomly chosen for experimental groups. Four- to six-week-old mixed gender AG129 mice were used in the immunization assays and ZIKV infection studies. The mice were randomly chosen for experimental groups. All mice were kept in a specific-pathogen-free facility at Yale University.

### 2.4. Plasmids and the Purification of Recombinant Proteins 

AgBR1 with a TEV tag was cloned in frame into the pMT-Bip-V5-His tag vector (Invitrogen) and recombinant proteins expressed and purified using the Drosophila Expression System (Invitrogen). AgBR1-TEV-V5-His was purified from the supernatant by TALON metal affinity resin (Clontech) and eluted with 150 mM imidazole. The eluted samples were filtered through a 0.22 μm filter and concentrated with a 10-kDa concentrator (Sigma-Aldrich) by centrifugation at 4 °C, and washed and dialyzed in PBS. Recombinant protein purities were assessed by SDS-PAGE, and then quantified using the Pierce BCA Protein Assay kit (Thermo Scientific). TurboTEV Protease (Accelagen Inc.) was used to cleave the tags from AgBR1-TEV-V5-His following the manufacturer’s instructions to obtain untagged AgBR1 protein for immunizations.

### 2.5. Immunoblotting

Purified protein was mixed in 1x Laemmli buffer (Biorad) with 2-mercaptoethanol (2.5%), heated to 95 °C for 10 min and separated by SDS-PAGE using 4%–20% Mini-Protean TGX gels (Bio-Rad) at 200 V for 25 min. Proteins were semi-dry transferred onto a polyvinylidene difluoride (PVDF) membranes (Millipore, Bedford, MA) for 45 min at 4 V. The blots were blocked in 1% non-fat milk in water for 60 min. AgBR1 and homologous proteins were incubated with AgBR1 antiserum (1:1000 dilution) for 1 h at room temperature or 4 °C overnight. Horseradish peroxidase-conjugated secondary antibodies were diluted in PBS-T and incubated for 1 h at room temperature (CST). After washing with 0.05% PBS-T, the immunoblots were imaged through ECL Western Blotting detection reagents (GE Healthcare) with a LI-COR Odyssey imaging system. 

### 2.6. Active Immunization Studies

For the active immunization assays, mice were immunized subcutaneously with 10-μg AgBR1 [16,17,18,19,20] with aluminum hydroxide (AH-AgBR1) or aluminum hydroxide alone (AH) boosted twice every two weeks with the same amount of AH-AgBR1 or AH (1:1 AH-AgBR1 or AH with sterilized PBS, 300ul per dose). Two weeks after the final immunization, the mice were anaesthetized with ketamine-xylazine and challenged by ZIKV-infected mosquitoes. For mosquito infection, ZIKV-filled needles were inserted into the thorax of each mosquito and 100 plaque forming units (PFU) in a volume of 138 nl were injected, using a Nanoject II Auto-Nanoliter Injector (Drummond). Infected mosquitoes were placed back in paper cups with mesh lids and maintained in triple containment for ten days in a warm chamber. The day before challenge, two infected mosquitoes were randomly selected and placed into individual cups with mesh covers, and, the following day, the mice were anaesthetized and ZIKV-infected mosquitoes were allowed to feed on them until observing blood engorgement. After challenge, mosquitoes were knocked down on ice and their thorax was dissected to examine the virus levels after mosquito feeding. To analyze viremia levels, the blood of the mosquito-bitten mice was collected at days 1, 3, 5, 7, and 9 post infection. Survival and weights were monitored daily. Mice exhibiting a weight loss of >20% of initial body weight or neurological disease were euthanized.

### 2.7. Quantitative Real Time PCR

The RNA from the thorax was extracted with the RNeasy Mini kit (QIAGEN) and the cDNA was generated with an iScript cDNA synthesis kit (Bio-Rad) according to the manufacturer’s protocol. Gene expression was examined by qRT-PCR using IQ SYBR Green Supermix. Viral RNA levels in mosquito thorax were normalized to *Rp49* RNA levels according to 2^−ΔCt^ calculations. Viral RNA levels in mouse blood were normalized to mice β-actin RNA levels according to 2^−ΔCt^ calculations.

### 2.8. ELISA

In total, 100 ng of recombinant AgBR1 were coated on Nunc-Immuno 96 MicroWell solid plates (Thermo Scientific) overnight at 4 °C. After being blocked with 2% non-fat milk for 1 h at 37 °C, the plates were incubated for 1 h 37 °C with serum samples serially diluted in PBS. After three washes with PBS+0.05% Tween20, the plates were incubated with horseradish peroxidase-conjugated secondary antibodies. Enzyme activity was detected by incubation with 50 μL of 3,3′,5,5′-tetramethylbenzidine solution (KPL) for 5 min at room temperature in the dark. The reaction was stopped by the addition of 50 μL of 1 M H_2_SO_4_. The optical density (OD) at 450 nm was measured with a microplate reader.

### 2.9. Statistical Analysis

GraphPad Prism software was used to perform statistical analysis on all data. The viral titers and change in the % of body weight were analyzed using the Wilcoxon–Mann–Whitney test. The survival rate was analyzed using a Log-rank test. *p*-values < 0.05 were considered statistically significant and are listed in the figure legends.

## 3. Results and Discussion

To obtain pure AgBR1 protein for vaccination studies, we expressed in Drosophila S2 cells a polyhistidine-tagged AgBR1 protein with a TEV recognition site located upstream of the His tag, using a pMT-Bip-V5-His vector (Thermo Fisher) and purified from the supernatant by TALON metal affinity resin (Clontech). The purified recombinant protein (Fig. 1a, left lane) was TEV protease-treated to cleave the V5-His tag (Figure 1a, right lane) [21]. A rabbit polyclonal antiserum raised against recombinant AgBR1-V5-His protein detected both AgBR1-TEV-V5-His and the cleaved version by Western Blotting (Figure 1b, left and right, respectively). To confirm complete TEV-cleavage, a His tag-specific antibody was also tested, and this antibody did not recognize the cleaved AgBR1 (Figure 1c).

We previously showed that AgBR1 antiserum suppresses inflammatory responses at the bite site and reduces lethality after ZIKV-infected mosquitoes fed on AG129 mice deficient in both interferon α and interferon γ receptors [14]. Therefore, we examined the protective effect of active immunization with AgBR1, recognizing that AG129 mice may not mount an optimal immune response. Alum Hydrogel was used as adjuvant because it is approved for human use [22]. We actively immunized two groups of mice: AgBR1 with Alum Hydrogel (AH-AgBR1) or Alum Hydrogel (AH) as control. 10 μg of purified AgBR1 protein was subcutaneously injected for immunization, following a prime–boost–boost regimen (Figure 2a). Blood samples were collected one day before priming and 13 days after final immunization to measure anti-AgBR1 antibody titer by ELISA. The anti-AgBR1 antibody titers in the AH-AgBR1 group were, on average, significantly higher than the titers in the AH group, (Figure 2b) even though AG129 mice are immunocompromised [23]. Three mice out of seven mounted a pronounced humoral response with high anti-AgBR1 antibody titers (AH-AgBR1-2, 4 and 5).

We next determined whether active immunization with AgBR1 protects mice against ZIKV transmission. A total of 100 PFU ZIKV virus were injected into the thorax of each mosquito using a Nanoject II Auto-Nanoliter Injector (Drummond). Then, two of the ZIKV-infected mosquitoes fed until engorgement on each mouse actively immunized with AH-AgBR1 or AH. There was no difference in ZIKV level in the thorax between the two groups of mosquitoes (Figure 3a). After mosquito transmission, we monitored mice to assess their survival and body weight loss for 30 days and viremia for 9 days. A statistically significant delay in mortality was observed in AH-AgBR1 group compared to AH control group. Between day 15 and 19 post-infection, more than 80% of AH-AgBR1 mice were alive, whereas fewer than 20% were alive in the AH control group (Figure 3b). We measured the viremia of these mice by quantitative reverse transcription-PCR (qRT-PCR), and on day 5 post-infection, the mice that were immunized with AH-AgBR1 showed lower viral titers than AH-immunized mice (Figure 3c). Moreover, there was a delay in body weight loss in AH-AgBR1 mice compared with AH mice (Figure 3d). On average, these differences in body weight are statistically significant at 10 day post-infection. We observed that the animals with the highest anti-AgBR1 antibody titers (AH-AgBR1-2, 4 and 5) had the most pronounced delay in mortality and body weight loss. Intriguingly, mice AH-AgBR1-3, 6, and 7 appear to be partially protected despite the lack of a robust humoral immune response, suggesting that other components of the adaptive immune system may contribute to antiviral protection (Table 1).

This study suggests that active immunization with AgBR1 adjuvanted with aluminum hydroxide delays lethal mosquito-borne ZIKV infection in AG129 mice. In addition, we did not observe inflammation at the site of inoculation, clinical signs, morbidity or any other appreciable pathological sign in the immunized mice, making this immunization strategy safe for mouse studies. The AG129 mouse model has been established as an adequate murine model to assess vaccine and antiviral efficacy against ZIKV [24]. This model also reproduces some aspects of the pathogenesis described in the natural host, as neuropathology, sexual, and vertical transmission [25,26]. As we were able to influence infection using an immunocompromised mouse (which is more susceptible to ZIKV), it is likely that more pronounced immune responses to AgBR1 will be obtained following the active immunization of immunocompetent mice or other animal models. Moreover, AG129 also have an inability to elicit robust host responses to ZIKV, which accounts for their high susceptibility to lethal infection. It is also possible that the delay in time to death following immunization with AgBR1, observed in our studies, may enable an immunocompetent mouse to develop a more robust direct response to the virus, which would aid in viral clearance and protection. 

The Asian lineage of ZIKV has recently caused epidemics and severe diseases, being responsible for the recent epidemics in the Americas in 2015 [27], leading to millions of human infections [28]. These strains have adapted to generate higher viremia in humans, leading to enhanced cross-placental infection and microcephaly neurological disease and weight loss [26]. Then, Asian strains, such as the Cambodia strain used in this work, could be a good tool to study the efficacy of the vaccine candidates. 

Moreover, ZIKV has been shown to be transmitted by different *Aedes* mosquito species, and two anthropophilic *Aedes* mosquitoes, *A. aegypti* and *A. albopictus*, are able to transmit ZIKV to humans [29]. As the sequences of *A. aegypti* and *A. albopictus* AgBR1 proteins share 92.3% identity, it is highly plausible that the antibodies developed against the *A. aegypti* AgBR1 can also block the *A. albopictus* homolog, thereby exerting an effect on ZIKV transmission. Therefore, targeting the mosquito salivary factor AgBR1 may direct the development of a new vaccine, either by blocking viral spread in the host or improving the efficacy of conventional vaccines that directly target the virus. 

## 4. Conclusions

In conclusion, this study presents a new vaccine candidate based on an *A. aegypti* salivary antigen AgBR1. An immunization regimen based on AgBR1 adjunvanted with the human-licensed alum hydroxide delayed the mortality rate in the highly ZIKV susceptible AG129 mouse model, preventing body weight loss and reducing viral burden, although the latter result should be examined carefully, since the viral burden analyzed by RT-PCR is a measurement of viral genome. Furthermore, analyzing efficacy in other less immunocompromised models, examining other adjuvants, ways of administration, or different delivery platforms, such as DNA and RNA-based vaccines, should be performed to achieve a maximal protection of AgBR1 candidate, and also could be potentially applied for other mosquito-based antigens.

## Figures and Tables

**Figure 1 vaccines-08-00145-f001:**
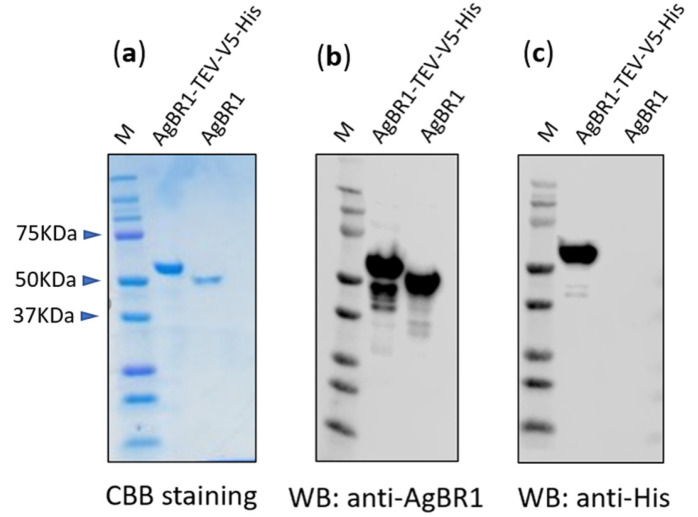
Expression of recombinant AgBR1 protein and polyhistidine-tag cleavage. Recombinant AgBR1 protein was expressed using stable transfected S2 cells, and the polyhistidine tag was cleaved by TEV protease. (**a**) Analysis of purified recombinant AgBR1-TEV-V5-His and His-cleaved AgBR1 protein by SDS-PAGE stained with Coomassie Brilliant Blue (CBB). (**b**) Analysis of AgBR1 His-cleaved recognition by rabbit anti-AgBR1 specific antibodies by Western Blotting. (**c**) Analysis of His-cleavage efficacy by Western Blotting using an anti-His tag specific antibody.

**Figure 2 vaccines-08-00145-f002:**
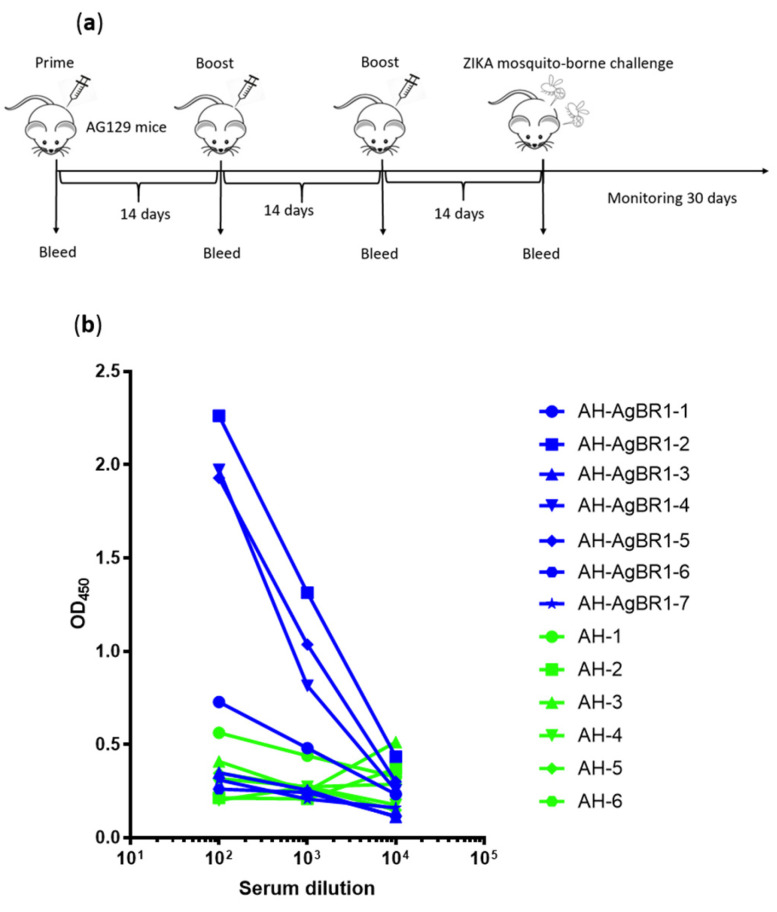
AG129 mice produce anti-AgBR1 antibodies after AgBR1 active immunization. 4-6 week old AG129 mice were immunized subcutaneously with 10 μg of AgBR1 adjuvanted with AlumHydro gel at the back base on weeks 0, 2, and 4 (AH-AgBR1, *n* = 7) or only AlumHydro gel served as a control (AH, *n* = 6), following a prime–boost–boost regimen. (**a**) Schematic of the experiment. Fourteen days after prime–boost–boost immunization, mice were challenged with Zika Virus (ZIKV) via mosquito bite and mice were monitored 30 days for survival, body weight, and viremia. (**b**) Mice antibody titer against AgBR1 was tested individually one day before challenge by ELISA (individual).

**Figure 3 vaccines-08-00145-f003:**
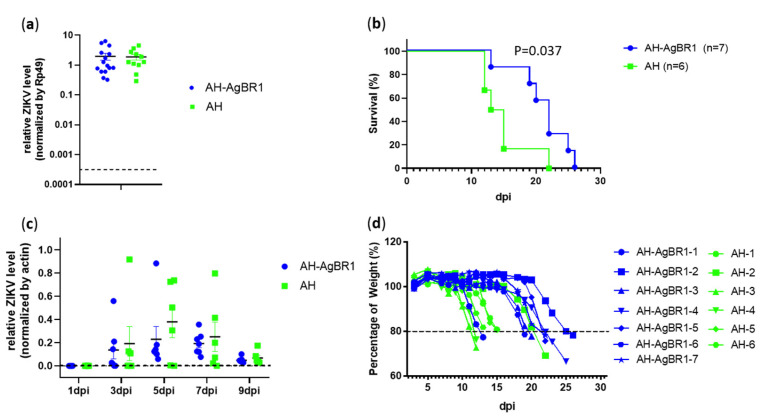
Active immunization in AgBR1 AG129 mice with AH-AgBR1 against mosquito-borne ZIKV infection. Survival rates, viremia, and body weight were analyzed in immunized AG129 mice after ZIKV infection. (**a**) The ZIKV viral load recovered from the thorax of ZIKV-infected *A. aegypti* mosquitoes after mice challenge. The basal line corresponds with the background observed in non-infected mosquitoes. ZIKV RNA levels were analyzed by qRT-PCR and normalized to the levels of *Rp49*. Each dot represents one mosquito. (**b**) The survival rates of AH-AgBR1 and AH mice after a mosquito-borne ZIKV challenge. The mice were observed every 24 h for 30 days. (**c**) The ZIKV viral load recovered from the blood of AH-AgBR1 and AH-immunized mice after challenge measured by qRT-PCR. Each point represents an individual animal. The relative ZIKV level was normalized to mouse β actin RNA levels. (**d**) The body weight loss (%) of mice after challenge. The error bars represent mean ± SEM. Asterisks represent significant difference between samples (AH-AgBR1 versus AH) for each analyzed day, calculated by Mann–Whitney non-parametric test (*p* ≤ 0.05). Survival statistical analysis was calculated by Log-Rank test (*p* ≤ 0.05). (Three separate experiments were operated biologically independently with the same results.)

**Table 1 vaccines-08-00145-t001:** AgBR1 antibody titers and the date of death post ZIKV infection, shown individually.

Mouse Number	AgBR1 Antibody Titiers (Serum Dilution: 1:100, OD_450_)	Date of Death (Day Post Infection)
AH		
AH-3	0.409	12
AH-4	0.200	12
AH-6	0.316	13
AH-5	0.346	15
AH-1	0.562	15
AH-2	0.213	22
AH-AgBR1		
AH-AgBR1-1	0.727	13
AH-AgBR1-6	0.260	19
AH-AgBR1-3	0.349	20
AH-AgBR1-7	0.310	22
AH-AgBR1-5	1.928	22
AH-AgBR1-4	1.971	25
AH-AgBR1-2	2.262	26

## Data Availability

Data that support the findings of this study are available from the corresponding authors upon request.

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
