# Peer review of "Vaccination with Aedes aegypti AgBR1 Delays Lethal Mosquito-Borne Zika Virus Infection in Mice"

_vaccines, 2020, doi:10.3390/vaccines8020145_

Round 1

Reviewer 1 Report

Overall comment - this manuscript suggests to me that this is the initial stage of a proof of concept and shows reasonable promise. However the differences between the immune systems of mice and humans are vast, as the authors will undoubtedly appreciate ?

Abstract: reword first sentence …. Asymptomatic infection, fever and headache … to asymptomatic infection, or acute infection with fever and headache …. Or asymptomatic and acute infections resulting in a variety of complications and sequelae.

Background; no changes

Material & Methods

Pg 3, line 106 why was 10ug AgBR1 chosen as the dosage ?

Results & Discussion

Pg 5, line 153 – Legend to Figure 1 first line seems incomplete ?

References adequate

General question: a number of other mosquito species, such as A. albopictus, africanus and others have been shown to transmit this virus. How effective would a vaccine using this salivary protein be in those regions where these other vectors are known to also transmit zikavirus ? I suggest that a sentence or two in the conclusion would be appropriate.

Author Response

Comments from the reviewers

Reviewer #1 (Remarks to the Author):

Overall comment - this manuscript suggests to me that this is the initial stage of a proof of concept and shows reasonable promise. However, the differences between the immune systems of mice and humans are vast, as the authors will undoubtedly appreciate?

This is a very important comment. We agree with reviewer that “the differences between the immune systems of mice and humans are vast”. Moreover, to best study the effect of AgBR1 on lethal ZIKV infection, we needed to use immunodeficient mice. Since these animals do not mount as robust an immune response as immunocompetent mice, it is likely that the differences we obtained would be more pronounced in animals and humans with a normal immune system. Therefore, the delay in mortality in the ZIKV sensitive AG129 mice, when immunized with AgBR1, serves as a proof of concept for the use of this mosquito salivary antigen as a vaccine candidate. We fully agree that studies in additional animal models should be performed to further develop this antigen as a human candidate for ZIKV prevention.

Abstract: reword first sentence …. Asymptomatic infection, fever and headache … to asymptomatic infection, or acute infection with fever and headache …. Or asymptomatic and acute infections resulting in a variety of complications and sequelae.

This change has been included in the text (Page 1, line 21).

Background; no changes

Material & Methods

Pg 3, line 106 why was 10ug AgBR1 chosen as the dosage?

Certainly, the assessment of different doses, as well as different adjuvants is important for future studies. There are several studies which show that 10 µg of antigen is common and sometimes optimal vaccine dose. In addition, 10 μg of protein in the vaccine formula would likely be cost-effective [1-6]. We have added the related references regarding the reason for dosage (Page 3, line 107).

Results & Discussion

Pg 5, line 153 – Legend to Figure 1 first line seems incomplete?

We have corrected the first line of Figure 1 legend (Page 5, line 150).

References adequate

General question: a number of other mosquito species, such as A. albopictus, africanus and others have been shown to transmit this virus. How effective would a vaccine using this salivary protein be in those regions where these other vectors are known to also transmit zika virus? I suggest that a sentence or two in the conclusion would be appropriate.

The reviewer is completely right. Other Aedes species have been shown to be vectors of ZIKV. The two anthropophilic Aedes mosquitoes that are able to transmit ZIKV are A.aegypti and A.albopictus [7]. We compared the sequences of both A.aegypti and A.albopictus AgBR1 proteins and we observed a 92.3% of identity between both. Based on this high similarity, it is likely that the antibodies developed against the A.aegypti AgBR1 may also block the A.albopictus homolog, and thereby influence ZIKV transmission. In addition, recent studies have shown a poor ability of A. albopictus to sustain a local transmission of ZIKV, suggesting that the influence of mosquito proteins may be even greater in this vector [8]. Following the reviewer’s suggestion, we have added this point in the discussion section (Page 7, line 236-240).

Reviewer 2 Report

Review for authors

This is an interesting paper on an AgBR1 enhance antibody production and examine and its influence on Zika virus infection. Manuscript is well written but to emphases the new vaccine of AgBR1, authors’ needs to improve including data presentation and methodology for publication in Vaccines.

Scientific comments,

#1. To analysis of Zika virus load in A. aegypti mosquitoes before the mice challenge, authors used the qRT-PCR for the ZIKV RNA levels. For assessment of vaccine efficacy, authors need to reconsider viral load for viremia in mice using TCID50 or PFU titration analysis.

#2. In this study, authors try to natural infection for the Zika virus into the mammalian host. However, still unclear that how much viruses were remained in A. aegypti mosquitoes after Zika virus injection and also how much viruses were infected in mice by virus infected mosquito.

#3. To emphases the new vaccine of AgBR1, author need to more clear interoperate that determined reason of the euthanasia guideline (weight loss of >20% of initial body weight) and need to more analysis of histopathological properties. In particular, only 3 of 7 were produced high anti-AgBR1 antibody titers and mice survival was not significantly different between vaccine and adjuvant only group which mean I guess delayed survival due to adjuvant effect than AgBR1 antigen.

Author Response

Reviewer #2 (Remarks to the Author):

Scientific comments,

#1. To analysis of Zika virus load in A. aegypti mosquitoes before the mice challenge, authors used the qRT-PCR for the ZIKV RNA levels. For assessment of vaccine efficacy, authors need to reconsider viral load for viremia in mice using TCID50 or PFU titration analysis.

We appreciate the reviewer’s comment and agree that in order to quantify infectious particles, a TCID50 or plaque assay is required. However, in our case, we are more focused on comparing the viral load between the control group and the AgBR1 immunized group to show that the immunized group has a reduction in the viral levels. In this work, we followed the same strategy to analyze viral burden as previous studies from our lab, which have shown that using the qRT-PCR is effective and feasible to analyze ZIKV viral load [9, 10].

#2. In this study, authors try to natural infection for the Zika virus into the mammalian host. However, still unclear that how much viruses were remained in A. aegypti mosquitoes after Zika virus injection and also how much viruses were infected in mice by virus infected mosquito.

The basal line in the Figure 3a showed the background of ZIKV genome in non-infected mosquitoes. In this figure, the viral burden of infected mosquitoes used to challenge AgBR1 and control mice are shown. We did not observe significant differences between them, but we did observe significant differences between infected and non-infected mosquitoes. In the legend we have now indicated that the basal line corresponds with the background observed in non-infected mosquitoes (Page 6, line 201-202). Regarding the amount of virus that is secreted within the saliva during mosquito bites, studies estimate that the amount of infectious particles varies from 3 to 4000 approximately in the case of Culex mosquito species and WNV [11], but other studies suggest that this number could be higher. For ZIKV in Aedes mosquitoes, 3 to 120 PFU have been found in the saliva [12]. However, these results have to be evaluated with care, since they were performed under experimental artificial infections. In addition, previous work performed in the lab showed the importance of using mosquito transmission in the mouse model to study the effect of mosquito secreted antigens on the host immune response and survival after ZIKV mosquito-borne challenge [9, 13].

#3. To emphases the new vaccine of AgBR1, author need to more clear interoperate that determined reason of the euthanasia guideline (weight loss of >20% of initial body weight) and need to more analysis of histopathological properties. In particular, only 3 of 7 were produced high anti-AgBR1 antibody titers and mice survival was not significantly different between vaccine and adjuvant only group which mean I guess delayed survival due to adjuvant effect than AgBR1 antigen.

The guidelines and policies followed in this work are under IACUC guidelines for animal research. Mice exhibiting a body weight loss of >20% of initial body weight should be euthanized under these guidelines, and often these animals shows neurologic disease as paralysis [13]. These are the university approved methods are appropriate for this study, and consistent with previous studies. We have survival, viremia, clinical signs and weight to evaluate vaccine efficacy. In addition, the pathology of ZIKV in the mouse model has been described [14]. We also made sure that the Zika viral load, the anti-AgBR1 antibody titers and the survival of the animals, were compared to the control group, which is adjuvant alone injected, at the same time. We also showed the viral load in a negative control with naive mice or mosquitoes. We agree with the reviewer about the point that only 3 to 7 mice displayed significant antibody titers against AgBR1 protein, but these mice were found to have a higher delay in mortality and body weight loss, showing significant differences between the immunized group and the control group only inoculated with adjuvant. We have more carefully stated these points in the text (Table 1).

Reviewer 3 Report

Vaccination with Aedes Aegypti AbBR-1 delays lethal mosquito-borne Zika virus infection in mice

This manuscript describes a study of Zika virus pathogenesis after Aedes Aegypti AbBR-1 vaccination of AG129 mice in an admixture with Alum hydrogel. The result of these vaccinations is a reduction in the time of death, with 100% mortality.

Overall comments

 While AG129 mice are used for Zika virus vaccine models, it is not clear why this model was chosen over the less severely compromised A129 mouse, especially since some discussion in the manuscript of efficacy in wild type mice suggests that this regimen would be equally effective. The selection of the Cambodian ZIKV strain is also not justified. This is an important issue because it is known that Asian ZIKV strains are more pathogenic to mice than African strains (1, 2). For these reasons, it is paramount for the study plan to include the appropriate virus-mouse model with the caveat that the observed pathogenic response must be interpreted with caution because it may not represent a human response. Is there an AgBR1 protein homolog in Aedes Albopictus, the alternate vector?

It is now routine to assay virus levels as relative RNA expression which is easy to do but difficult to interpret and compare to the data from established flavivirus models which report virus titer as pfu/mL or ffu/mL. It would be extremely helpful if the investigator were to quantify how the RNA levels compare to pfu/mL.

Because the data vary so much it would be helpful if data for the post challenge response and the survival rates were reported individually in a table.

The level of mortality in the AG129 mice is reduced in time (5 days) and requires 3 administrations of the vaccine. This vaccine does not appear to be efficacious.  Also as a practical matter, because for this strategy to be applied to humans this vaccine would require multiple boosts, compliance with additional inoculations would be low.  The ideal vaccine would not require a boost, only one injection; second choice would be a single boost. No data from fewer injections is presented or discussed. Another issue is that the dose is high at 10 µg/injection thus the efficacy of this vaccine is low, even when adjuvanted.

Not addressed is the safety of this type of vaccine. Humans have antibodies to mosquito antigens and do not represent a naïve population. Safety was not evaluated and is a critical part of any vaccine strategy intended for humans or animals. Mice expressing salivary protein antibodies should be immunized with the AgBR1 protein and then challenged with the virus. Very low levels of mosquito antigens have been known to cause anaphylaxis in humans.

Without additional control and experiments to justify this platform this vaccine is weak at best. The efficacy of this virus is low and safety is not addressed. Without more and better arguments to move this vaccine forward and a discussion of next steps the methodology is adequate but the argument that it will serve as an effective addition to a human vaccine against arbovirus disease is very weak.

  1. Tripathi S, Balasubramaniam VRMT, Brown JA, Mena I, Grant A, Bardina SV, Maringer K, Schwarz MC, Maestre AM, Sourisseau M, Albrecht RA, Krammer F, Evans MJ, Fernandez-Sesma A, Lim JK, García-Sastre A. 2017. A novel Zika virus mouse model reveals strain specific differences in virus pathogenesis and host inflammatory immune responses. PLOS Pathogens 13:e1006258.
  2. Morrison TE, Diamond MS. 2017. Animal Models of Zika Virus Infection, Pathogenesis, and Immunity. J Virol 91.

Author Response

Reviewer #3 (Remarks to the Author):

Overall comments

While AG129 mice are used for Zika virus vaccine models, it is not clear why this model was chosen over the less severely compromised A129 mouse, especially since some discussion in the manuscript of efficacy in wild type mice suggests that this regimen would be equally effective. The selection of the Cambodian ZIKV strain is also not justified. This is an important issue because it is known that Asian ZIKV strains are more pathogenic to mice than African strains (1, 2). For these reasons, it is paramount for the study plan to include the appropriate virus-mouse model with the caveat that the observed pathogenic response must be interpreted with caution because it may not represent a human response. Is there an AgBR1 protein homolog in Aedes Albopictus, the alternate vector?

It is now routine to assay virus levels as relative RNA expression which is easy to do but difficult to interpret and compare to the data from established flavivirus models which report virus titer as pfu/mL or pfu/mL. It would be extremely helpful if the investigator were to quantify how the RNA levels compare to pfu/mL.

We agree that this is important and have addressed this in a previous comment.

Because the data vary so much it would be helpful if data for the post challenge response and the survival rates were reported individually in a table.

As reviewer suggested, we have added the information about the mice post challenge response individually in a table, to facilitate the comprehension of the results (Page 6, line 212).

The level of mortality in the AG129 mice is reduced in time (5 days) and requires 3 administrations of the vaccine. This vaccine does not appear to be efficacious.  Also, as a practical matter, because for this strategy to be applied to humans this vaccine would require multiple boosts, compliance with additional inoculations would be low.  The ideal vaccine would not require a boost, only one injection; second choice would be a single boost. No data from fewer injections is presented or discussed. Another issue is that the dose is high at 10 µg/injection thus the efficacy of this vaccine is low, even when adjuvanted.

The essential goal of vaccination is to generate potent and long-term protection against diseases. Previous studies have shown that prime-boost vaccine strategies could enhance cellular and also humoral immunity in several animal models among different vaccine modalities [21]. The AG129 mice lack both IFN−α/β and −γ receptors, but elicit B-cell and T-cell responses to infection [22]. As Type I interferon signaling in B- and CD4+ T-cells is required for optimal antibody response, vaccine studies in this mouse model do not provide a full measure of immune correlates of protection. Nevertheless, it is an effective animal model to study vaccine efficacy against viremia, disease pathogenesis and mortality [15]. Antibody titer levels after first boost were still low in the immunized animals. Given this fact, the prime-boost-boost strategy followed in this work was required to enhance the humoral immune response against the mosquito antigen AgBR1. In a less immunodeficient mouse model (IFNAR) and in immunocompetent models we would expect to achieve enough level of antibody response with lesser number of doses. Regarding the amount of protein used to immunize animals, there are several studies which show that 10 µg of antigen is optimum for a vaccine dose and even more, and 10 μg of protein in the vaccine formula would be cost-effective [1-6].

We agree with the reviewer that the ideal vaccine would not require a boost, only one injection. Future studies in animals and humans will hopefully address this issue.

Not addressed is the safety of this type of vaccine. Humans have antibodies to mosquito antigens and do not represent a naïve population. Safety was not evaluated and is a critical part of any vaccine strategy intended for humans or animals. Mice expressing salivary protein antibodies should be immunized with the AgBR1 protein and then challenged with the virus. Very low levels of mosquito antigens have been known to cause anaphylaxis in humans.

We thank the reviewer for this comment. Alum Hydroxide is a human-licensed adjuvant. Mice immunized with AgBR1 antigen adjuvanted with alum did not show inflammation at the site of inoculation, clinical signs, morbidity or any other appreciable pathological sign.  We agree with reviewer that human populations are exposed to mosquito bites, therefore developing antibodies against mosquito antigens. However, the antibody titer against AgBR1 protein displayed after mosquito bites in humans is not known, and also it has been described that some mosquito antigens do not induce strong antibody titers after mosquito bites [23], although they contribute to the spread of some pathogens after transmission, and antibodies directed against this antigen restrict the course of the infection [24].

The typical clinical course of sensitization and natural desensitization to mosquito salivary allergens was described initially in the 1940s. We only know a specific case in which a mosquito bitten person develop anaphylaxis with a systemic mastocytosis [25]. People never exposed to a particular species of mosquito do not develop reactions to the initial bites and, subsequent bites result in the appearance of delayed local skin reactions. After repeated bites, immediate wheals appear and, with further exposure, the delayed local reactions wane and eventually disappear, although the immediate reactions persist. People who are repeatedly exposed to bites from the same species of mosquito eventually also lose their immediate reactions. These typical reactions are annoying but not dangerous. Based on this, we consider that the risk of anaphylaxis shock that this vaccine candidate could generate is very low.  However, we have modified the text to indicate that the safety of an AgBR1-base vaccine must be carefully assessed in animals and humans (Page 7, line 215-217).

Without additional control and experiments to justify this platform this vaccine is weak at best. The efficacy of this virus is low and safety is not addressed. Without more and better arguments to move this vaccine forward and a discussion of next steps the methodology is adequate but the argument that it will serve as an effective addition to a human vaccine against arbovirus disease is very weak.

We agree with reviewer that the vaccine strategy described in this work can be improved - indeed this is the case with virtually all vaccines and vaccine candidates. Examining other adjuvants, doses of vaccine, other animal models and other methods of administration such as DNA and RNA based vaccines, will all contribute to our understanding of the maximal protective effect of AgBR1 and potentially other mosquito antigens. The text has been modified to make this clearer (Page 7, line 241-246).

  1. Tripathi S, Balasubramaniam VRMT, Brown JA, Mena I, Grant A, Bardina SV, Maringer K, Schwarz MC, Maestre AM, Sourisseau M, Albrecht RA, Krammer F, Evans MJ, Fernandez-Sesma A, Lim JK, García-Sastre A. 2017. A novel Zika virus mouse model reveals strain specific differences in virus pathogenesis and host inflammatory immune responses. PLOS Pathogens 13: e1006258.
  2. Morrison TE, Diamond MS. 2017. Animal Models of Zika Virus Infection, Pathogenesis, and Immunity. J Virol 91.

We are grateful to the editor and reviewers for the constructive comments to help improve the manuscript. We hope it is now suitable for publication.

Round 2

Reviewer 2 Report

I appreciate the care the authors took with revising the manuscript.

Author Response

We really appreciate the suggestions kindly provided by reviewer #2 in the first round of revisions.

Reviewer 3 Report

C6/36 cells are Aedes Albopictus derived.

Rationale for lack of some type of direct infectivity assay is weak. There needs to be a pfu/mL ffu/mL or TDID50 measurement to evaluated the virus load. The particle to pfu ratio of these viruses can vary widely and the level of RNA is not an indicator of infectious virus which at some point will be important in the development of any vaccine. This report should include some type of infectivity data.

Author Response

Reviewer #3 (Remarks to the Author):

C6/36 cells are Aedes albopictus derived.

As the reviewer indicated, C6/36 cells were generated from A. albopictus embryos. This is a well-established mosquito cell line, widely used in flaviviral research. The viral titer obtained using this cell line is often higher than in other cell lines such as Aag2 (from A. aegypti). This is because the C6/36 cells contain deletions in the RNA interference system, and therefore support higher replication of the virus [1].

Rationale for lack of some type of direct infectivity assay is weak. There needs to be a pfu/mL ffu/mL or TDID50 measurement to evaluate the virus load. The particle to pfu ratio of these viruses can vary widely and the level of RNA is not an indicator of infectious virus which at some point will be important in the development of any vaccine. This report should include some type of infectivity data.

The reviewer makes an important point.  In the context of an infection being controlled by the immune response, the ratio of live versus dead viruses can vary.  PCR data will show all “dead and alive” virus, while an infectivity measurement would focus on the live virus (infectious particles). In addition, there are some studies that correlate viral burdens measured by PCR and the infectious particle measured by plaque assay [2, 3]. Culture results would not affect the conclusions of the paper. We completely agree with the reviewer that an assay to measure infectious particles would help demonstrate whether sterile immunity was associated with survival.  However, we are reporting a delay in time to death of the experimental compared to the control animals. To reinforce this, we analyzed an additional marker of infection and observed a delay in the loss of body weight in the immunized mice. These points are more clearly stated. In addition, the use of PCR to detect viral genomes, and not to distinguish live or dead virus, is now detailed in the discussion (243-245).

  1. Brackney DE, Scott JC, Sagawa F, et al. C6/36 Aedes albopictus cells have a dysfunctional antiviral RNA interference response. PLoS Negl Trop Dis 2010; 4:e856.
  2. Bae HG, Nitsche A, Teichmann A, Biel SS, Niedrig M. Detection of yellow fever virus: a comparison of quantitative real-time PCR and plaque assay. J Virol Methods 2003; 110:185-91.
  3. Perkins SM, Webb DL, Torrance SA, et al. Comparison of a real-time reverse transcriptase PCR assay and a culture technique for quantitative assessment of viral load in children naturally infected with respiratory syncytial virus. J Clin Microbiol 2005; 43:2356-62.